# Cardiopulmonary Bypass-Induced IL-17A Aggravates Caspase-12-Dependent Neuronal Apoptosis Through the Act1-IRE1-JNK1 Pathway

**DOI:** 10.3390/biom15081134

**Published:** 2025-08-06

**Authors:** Ruixue Zhao, Yajun Ma, Shujuan Li, Junfa Li

**Affiliations:** 1Department of Neurobiology, School of Basic Medical Science, Capital Medical University, Beijing 100069, China; ruixuezhao@mail.ccmu.edu.cn; 2The Neurological Department, Fu Wai Hospital, National Center for Cardiovascular Diseases, Chinese Academy of Medical Science and Peking Union Medical College, Beijing 100037, China; mayajun@fuwai.com

**Keywords:** cardiopulmonary bypass, IL-17A, endoplasmic reticulum stress, neuronal apoptosis

## Abstract

Cardiopulmonary bypass (CPB) is associated with significant neurological complications, yet the mechanisms underlying brain injury remain unclear. This study investigated the role of interleukin-17A (IL-17A) in exacerbating CPB-induced neuronal apoptosis and identified vulnerable brain regions. Utilizing a rat CPB model and an oxygen–glucose deprivation/reoxygenation (OGD/R) cellular model, we demonstrated that IL-17A levels were markedly elevated in the hippocampus post-CPB, correlating with endoplasmic reticulum stress (ERS)-mediated apoptosis. Transcriptomic analysis revealed the enrichment of IL-17 signaling and apoptosis-related pathways. IL-17A-Neutralizing monoclonal antibody (mAb) and the ERS inhibitor 4-phenylbutyric acid (4-PBA) significantly attenuated neurological deficits and hippocampal neuronal damage. Mechanistically, IL-17A activated the Act1-IRE1-JNK1 axis, wherein heat shock protein 90 (Hsp90) competitively regulated Act1-IRE1 interactions. Co-immunoprecipitation confirmed the enhanced Hsp90-Act1 binding post-CPB, promoting IRE1 phosphorylation and downstream caspase-12 activation. In vitro, IL-17A exacerbated OGD/R-induced apoptosis via IRE1-JNK1 signaling, reversible by IRE1 inhibition. These findings identify the hippocampus as a key vulnerable region and delineate a novel IL-17A/Act1-IRE1-JNK1 pathway driving ERS-dependent apoptosis. Targeting IL-17A or Hsp90-mediated chaperone switching represents a promising therapeutic strategy for CPB-associated neuroprotection. This study provides critical insights into the molecular crosstalk between systemic inflammation and neuronal stress responses during cardiac surgery.

## 1. Introduction

Cardiopulmonary bypass (CPB) is a cornerstone of modern cardiac surgery, enabling life-saving interventions by maintaining systemic circulation and oxygenation. Despite its critical role in improving postoperative survival rates, CPB is associated with significant complications, among which brain injury is particularly concerning [1]. Neurological complications such as stroke, postoperative delirium, and cognitive dysfunction impose a substantial burden on both society and affected families [2]. Understanding the mechanisms underlying CPB-induced brain injury and identifying potential targets for neuroprotective interventions are therefore urgent priorities. However, the specific vulnerable brain regions and the molecular mechanisms driving CPB-related brain injury remain poorly understood.

During CPB, surgical trauma and exposure to artificial bypass materials can activate the immune system, triggering a systemic inflammatory response [3,4,5]. This inflammatory cascade elevates the levels of interleukin-17A (IL-17A), which has been implicated in exacerbating neuronal apoptosis through mechanisms involving endoplasmic reticulum stress (ERS) and caspase activation [6]. The ERS apoptosis pathway is one of the three classical apoptosis pathways. ERS is an important factor leading to neuronal ischemia/reperfusion (I/R) injury [7]. When ERS is too strong or persistent, apoptosis-related factors such as C/EBP (CCAAT enhancer binding protein) homologous protein (CHOP) and Caspase-12 are activated, triggering apoptosis [8,9]. The ERS is controlled by three main sensors: inositol-requiring enzyme 1 (IRE1), PKR-like ER kinase (PKRK), and Activating Transcription Factor 6 (ATF6) [10,11,12]. Among these, IRE1 is a transmembrane protein with both kinase and endonuclease (RNase) activities. Under the ERS condition, the ER domain of IRE1 can induce the unfolded protein to dissociate from the protein bound to its phosphokinase domain, thereby releasing its phosphokinase domain and forming a dimer through autophosphorylation, thus activating its RNase activity [13,14,15]. The activation of IRE1 can promote the increase in phosphorylation of JNK1, thereby regulating the expression of corresponding target genes and inducing pro-survival and pro-apoptotic cell reactions [16,17].

IL-17A exerts its biological effects by binding to a heterodimeric receptor complex composed of IL-17RA and IL-17RC, which initiates downstream signaling cascades [18]. A key mediator in this pathway is Act1, which interacts with IL-17R through its SEFIR domain to activate downstream signaling pathways, including NF-κB, MAPK, and C/EBP [18]. Notably, Act1 requires the molecular chaperone heat shock protein 90 (Hsp90) to maintain its structural integrity and functional activity [19]. Chenhui et al. [20] demonstrated the interaction between Hsp90 and Act1 using mass spectrometry and co-immunoprecipitation. Their findings revealed that Hsp90 inhibitors significantly attenuate Act1 signaling, and IL-17 stimulation enhances the interaction between Act1 and Hsp90. Therefore, Hsp90 plays a key role in IL-17A signal transduction. Furthermore, Hsp90 has been shown to interact with the kinase domain of IRE1 through its co-chaperone CDC37, modulating IRE1 autophosphorylation. Inhibitors of Hsp90, such as geldanamycin (GA), can induce IRE1 phosphorylation and oligomerization. Thus, Hsp90 serves as a pivotal regulator of both upstream Act1 and downstream IRE1, playing a key bridging role in IL-17A signal transduction.

In this study, transcriptomic analysis was employed to identify differentially expressed genes (DEGs) in rat brain tissue following CPB. Additionally, a rat CPB model and an in vitro 1 h oxygen–glucose deprivation/24 h reoxygenation (OGD/R) model were utilized to investigate the vulnerable brain regions after CPB and the molecular mechanism of IL-17A aggravating brain injury. The neuroprotective potential of IL-17A-Neutralizing mAb after CPB was confirmed by exogenous administration, which provided a new target for brain protection intervention during CPB.

## 2. Materials and Methods

### 2.1. Animals

Male Sprague Dawley (SD) rats (weighted 500 ± 50 g, age 12–14 weeks, purchased from the HFK Bioscience, Beijing, China) were selected and randomly divided into 5 groups, with *n* = 6 in each group: the Sham group, the CPB group, the CPB + IgG isotype group, the CPB + IL-17A Ab group, and the CPB + 4-PBA group. The experiment received support from the Institutional Animal Care and Use Committee (IACUC), Fuwai Hospital, Chinese Academy of Medical Sciences [0108-7-90-ZX (X)].

### 2.2. Model of CPB

Anesthesia was induced in rats via inhalation of 3% isoflurane followed by endotracheal intubation using a 16G arterial cannula. Mechanical ventilation was initiated (tidal volume: 8 mL/kg; respiratory rate: 80 breaths/min) using a rodent ventilator. The left femoral artery was surgically exposed and cannulated with a 24G arterial catheter connected to a multi-channel physiological monitoring system for continuous arterial blood pressure and heart rate recording. Concurrently, the caudal artery was isolated and cannulated with a 20G arterial catheter to serve as the arterial inflow line for the CPB circuit. The right jugular vein was exposed, and a custom-designed multi-orifice venous drainage catheter was advanced into the right atrium via this access. Systemic heparinization (500 IU/kg) was administered prior to CPB initiation. A closed-loop extracorporeal circulation system was established, comprising gravity-dependent venous drainage into a reservoir and subsequent propulsion via a roller pump into a membrane oxygenator. The extracorporeal circulation system employed a well-established, commercially available membrane oxygenator specifically designed for small animals (XIJIAN MEDICAL, Beijing, China), featuring a polypropylene hollow-fiber membrane with a pore size of 0.03 μm to minimize inflammatory activation. During CPB, blood flow was maintained at 160–180 mL/kg/min under 1.5% isoflurane anesthesia. Core temperature was regulated using a rodent-specific warming pad throughout the 2 h CPB period. Sham controls received equivalent arterial/venous cannulation without circuit activation (Figure 1a).

For terminal tissue collection, rats were euthanized via rapid exsanguination under deep anesthesia. Brains were immediately harvested, sagittally bisected, and either fixed in 4% paraformaldehyde/glutaraldehyde or flash-frozen in liquid nitrogen for long-term storage at −80 °C.

In survival cohorts, vascular access sites were ligated following CPB discontinuation. Muscular and cutaneous layers were closed sequentially. After ventilator weaning and tracheal extubation, animals were monitored for 12 h prior to neurological assessment using modified Neurological Severity Scores (mNSS). Post-evaluation euthanasia was performed as described above.

### 2.3. Drug Treatment

Rats in the CPB + IL-17A Ab group received intravenous administration of an IL-17A-Neutralizing monoclonal antibody (mAb; clone eBio64DEC17, eBioscience, San Diego, CA, USA) via the jugular vein 30 min prior to CPB initiation, at a dosage of 100 μg/kg body weight. The CPB + IgG isotype group was injected with an equivalent volume of species-matched IgG isotype control (same administration route and timing). For the CPB + 4-PBA group, fully referring to previous research experience [21,22], the rats were subjected to daily intraperitoneal injections of the ERS inhibitor 4-phenylbutyric acid (4-PBA; No. HY-A0281, MedChemExpress, Monmouth Junction, NJ, USA) at 100 mg/kg/day for 14 consecutive days preceding CPB surgery.

### 2.4. Transcriptome Sequencing

Cerebral tissues were harvested from five randomly selected rats per group (the Sham group and the CPB group) using pre-chilled surgical instruments. The entire extraction procedure was conducted on ice with strict adherence to a 5 min ischemic window to minimize post-mortem molecular alterations. Samples are then sent to the Shanghai Applied Protein Technology Co., Ltd. (Shanghai, China) for transcriptomic sequencing to analyze the DEGs. The DEGs were acquired with |log_2_ (FC)| > 1 and *p* (adj.) < 0.05. The DEGs were analyzed by using the Kyoto Encyclopedia of Genes and Genomes (KEGG) annotation.

### 2.5. In Vitro OGD/R Model

The rat neuroblastoma cell line B104 (CL-0784, Pricella, Wuhan, China) was employed to establish the OGD/R model. Previous studies have confirmed that 2% O_2_ can induce mitochondrial dysfunction and oxidative stress in neurons, which is consistent with the mechanism of in vivo ischemic injury [6,23]. Compared with lower oxygen concentrations (e.g., 0.1% or 1% O_2_), 2% O_2_ can be more stably controlled by conventional tri-gas incubators, reducing experimental errors caused by fluctuations in oxygen concentration. Therefore, in order to balance the pathophysiological significance and the reproducibility of the experiment, 2% O_2_ was selected for OGD treatment in this experiment. For OGD induction, cultures were transferred to glucose-free DMEM and placed in a tri-gas incubator pre-equilibrated to 2% O_2_, 5% CO_2_, and 93% N_2_ for 1 h. Reperfusion was initiated by replacing the medium with complete growth medium (DMEM + 10% fetal bovine serum) under normoxic conditions (21% O_2_, 5% CO_2_, 74% N_2_) for 24 h. Pharmacological interventions—recombinant IL-17A (250 ng/mL; RP-8621, Thermo Fisher Scientific, Waltham, MA, USA) and the IRE1α-specific inhibitor GSK2850163 (HY-U00459, MedChemExpress, Monmouth Junction, NJ, USA)—were administered at OGD initiation.

### 2.6. Enzyme-Linked Immunoassay (ELISA)

Brain tissues isolated from distinct rat neuroanatomical regions were homogenized in ice-cold physiological saline (0.9% NaCl) at a specific tissue-to-buffer ratio. Homogenates were centrifuged at 12,000× *g* for 10 min under refrigeration (4 °C), followed by careful collection of supernatants while avoiding pellet contamination. IL-17A concentrations in clarified supernatants were quantified using a commercial ELISA kit (EK317/3-48, MULTI SCIENCES, Hangzhou, China) following the manufacturer’s standardized protocol.

### 2.7. Western Blotting

Tissue or cellular samples were lysed in RIPA buffer (P0013B, Beyotime Biotechnology, Shanghai, China) supplemented with protease and phosphatase inhibitors (cOmplete™ Mini and PhosSTOP™, Roche, Basel, Switzerland). Protein concentrations were quantified using a BCA assay kit (P0011, Beyotime Biotechnology, Shanghai, China). Protein lysates were separated by using 4–12% SDS-PAGE (NP0322BOX, Invitrogen, Carlsbad, CA, USA) and electrotransferred onto nitrocellulose membranes (IB33001, Invitrogen, Carlsbad, CA, USA). Membranes were blocked with 5% non-fat milk for 90 min at room temperature, followed by overnight incubation with primary antibodies at 4 °C: anti-caspase-3 (1:1000; 19677-1-AP, Proteintech, Wuhan, China), anti-caspase-8 (1:1000; 13423-1-AP, Proteintech, Wuhan, China), anti-caspase-9 (1:1000; 10380-1-AP, Proteintech, Wuhan, China), anti-caspase-12 (1:1000; ab62484, Abcam, Cambridge, UK), anti-Bax (1:2000; 50599-2-Ig, Proteintech, Wuhan, China), anti-Bcl-2 (1:1000; 26593-1-AP, Proteintech, Wuhan, China), anti-IL-17RA (1:1000; 144-10052-50, RayBiotech, Atlanta, GA, USA), anti-Na+/K+-ATPase (1:5000; 14418-1-AP, Proteintech, Wuhan, China), anti-GRP78 (1:2000; 11587-1-AP, Proteintech, Wuhan, China), anti-CHOP (1:1000; 15204-1-AP, Proteintech, Wuhan, China), anti-IRE1 (1:1000; WL02562, WANLEIBIO, Shenyang, China), anti-p-IRE1 (1:1000; WL05299, WANLEIBIO, Shenyang, China), anti-PERK (1:1000; WL03378, WANLEIBIO, Shenyang, China), anti-p-PERK (1:1000; WL05295, WANLEIBIO, Shenyang, China), anti-ATF6 (1:2000; 24169-1-AP, Proteintech, Wuhan, China), anti-JNK1 (1:1000; WL05246, WANLEIBIO, Shenyang, China), anti-p-JNK1 (1:1000; WL01813, WANLEIBIO, Shenyang, China), anti-Hsp90 (1:2000; 13171-1-AP, Proteintech, Wuhan, China), anti-Act1(1:200; sc-398161, Santa Cruz Biotechnology, Santa Cruz, CA, USA), and anti-β-Tubulin (1:5000; 10094-1-AP, Proteintech, Wuhan, China). Then, membranes were incubated with HRP-conjugated species-specific secondary antibodies (rabbit or mouse IgG) for 1 h at room temperature. Protein bands were visualized using the Tanon 5800 Multi chemiluminescence imaging system (Tanon Science & Technology, Shanghai, China) and quantified via ImageJ software version 1.8.0 (National Institutes of Health, Bethesda, MD, USA). The levels of proteins in blots were calculated as a ratio of the grayscale value of the target protein to the grayscale value of the internal reference protein in the respective lane of the blot and defined as “Ratio” in all figures.

### 2.8. Hematoxylin–Eosin (H&E) Staining and Nissl Staining

Following extraction, rat brain tissues were fixed in 4% paraformaldehyde at 4 °C for a minimum of 24 h, followed by paraffin embedding. Serial coronal sections (4 μm thick) of entire hippocampi were collected using a microtome. The sections were mounted on poly-L-lysine-coated slides and processed for both H&E and Nissl staining protocols. Histologic images were acquired using an upright light microscope (NIKON ECLIPSE E100, Nikon, Tokyo, Japan) and analyzed with ImageJ software to quantify the total number of damaged neurons in each section.

### 2.9. Co-Immunoprecipitation (Co-IP)

The Immunoprecipitation Kit with Protein A+G Magnetic Beads (P2179M, Beyotime Biotechnology, Shanghai, China) was used in accordance with the instructions. Cellular or tissue samples were lysed in protease inhibitor-supplemented lysis buffer, followed by centrifugation to collect supernatants, with total lysate (Input) retained for downstream analysis. Magnetic beads were incubated with specific antibodies or normal IgG controls at room temperature for 60 min to form bead–antibody complexes, followed by three washes with Tris-buffered saline (TBS) to remove unbound antibodies. The antibody-conjugated beads were then incubated with clarified lysates overnight at 4 °C on a rotary mixer to capture target protein complexes. Non-specific binding was minimized by washing the complexes three times with lysis buffer containing inhibitors. Finally, the captured protein complexes were eluted by adding SDS-PAGE Sample Loading Buffer (1×) and heating at 95 °C for 5 min, yielding denatured samples for subsequent electrophoresis analysis.

### 2.10. TUNEL Assay

Cellular apoptosis was assessed using a TUNEL assay kit (C1086, Beyotime Biotechnology, Shanghai, China). Following the removal of culture medium, cells were washed three times with phosphate-buffered saline (PBS, pH 7.4). Fixation was performed using immunofluorescence fixative (P0098, Beyotime Biotechnology, Shanghai, China) for 30 min at room temperature, followed by additional PBS washes. Cells were permeabilized with immunostaining permeabilization buffer (P0097, Beyotime Biotechnology, Shanghai, China) for 5 min at room temperature and subsequently washed with PBS. The pre-prepared TUNEL detection solution was applied, and samples were incubated at 37 °C for 60 min under light-protected conditions. Finally, slides were mounted using DAPI-containing mounting medium and imaged under a fluorescence microscope.

### 2.11. Statistical Analysis

Quantitative data are presented as mean ± standard error of the mean (SEM). Statistical significance was determined using either one-way or two-way analysis of variance (ANOVA), followed by Bonferroni post hoc tests for all pairwise multiple comparisons. Data collection and statistical analyses were performed using GraphPad Prism version 9.0 (GraphPad Software, San Diego, CA, USA). Statistical significance was defined as *p* < 0.05.

## 3. Results

### 3.1. DEGs Were Significantly Enriched in IL-17 Signaling and Apoptosis-Related Pathways, and IL-17A-Neutralizing mAb and 4-PBA Could Significantly Improve Neurological Dysfunction in Rats After CPB

To explore the intrinsic mechanisms underlying cerebral injury following CPB, we optimized and validated a rat CPB model (Figure 1a) based on previous methodologies, with experimental groups detailed in the Materials and Methods Section. In the Sham and CPB groups, five rats were randomly selected from each group for transcriptomic sequencing. Volcano plots (Figure 1c) were generated using *p*-values and fold-change thresholds to visualize differential gene expression, revealing 1377 DEGs between groups, including 942 upregulated and 435 downregulated genes (Figure 1d). Hierarchical clustering analysis intuitively demonstrated the expression patterns of genes across different samples, revealing distinct clustering between the Sham and CPB groups (Figure 1e). KEGG enrichment analysis identified DEGs enriched in critical pathways, including the IL-17 signaling pathway and apoptosis-related pathways (Figure 1f).

Survival modeling confirmed that the administration of an IL-17A-Neutralizing mAb and 4-PBA significantly attenuated neurological deficits in CPB rats, as evidenced by reduced mNSS at 24 h post-CPB (Figure 1b).

### 3.2. The Levels of IL-17A and Neuronal Apoptosis Were Significantly Increased in Hippocampi of Rats After CPB

To investigate the vulnerable brain regions in which IL-17A exacerbates neuronal apoptosis following CPB, IL-17A levels were quantified in the cortex, hippocampus, striatum, and thalamus of rats using ELISA. The results demonstrated a significant increase in IL-17A levels in the cortex (Figure 2a) and hippocampus (Figure 2b) post-CPB, whereas no significant changes were observed in the striatum (Figure 2c) or thalamus (Figure 2d). Notably, the hippocampus exhibited the most pronounced elevation in IL-17A levels. Cleaved-Caspase-3 levels, a marker of apoptosis, were assessed across the four brain regions (Figure 2e–l). Only the hippocampus (Figure 2g,h) showed a significant increase in Cleaved-Caspase-3 levels post-CPB, indicating heightened apoptotic activity. This increase was significantly attenuated by IL-17A-Neutralizing mAb treatment. No significant differences were observed between the IgG isotype control and CPB groups, ruling out nonspecific antibody effects. These findings confirm that CPB-induced IL-17A exacerbates apoptotic injury predominantly in the hippocampal region, with greater severity compared to other brain regions.

### 3.3. IL-17A Aggravated the CPB-Induced Hippocampal Injury, Which Could Be Significantly Reversed by IL-17A-Neutralizing mAb

To observe the pathological damage in the hippocampi of rats following CPB, we performed H&E staining (Figure 3a,c) and Nissl staining (Figure 3b,d) on the hippocampal brain tissues. The morphological features of damaged cells include pyknosis or karyorrhexis, cytoplasmic eosinophilia, Nissl body dissolution, and blurred cellular contours. Only cells with clearly identifiable normal nuclear and cytoplasmic structures were classified as normal neurons. The results revealed that, compared with the Sham group, the CPB group exhibited significantly more pronounced damage in the hippocampus, with a marked increase in the number of damaged cells. Treatment with IL-17A-Neutralizing mAb partially reversed the hippocampal injury induced by CPB.

Additionally, we assessed the membrane translocation level of IL-17RA in the hippocampus of rats after CPB (Figure 3e,f). The membrane translocation level of IL-17RA was calculated as follows: the ratio of the grayscale value of IL-17RA membrane protein to that of Na^+^-K^+^-ATPase was defined as A, and the ratio of the grayscale value of IL-17RA cytoplasmic protein to that of β-Tubulin was defined as B. The membrane translocation level of IL-17RA was then calculated as A/(A + B). Our results showed that this level was significantly increased after CPB, indicating that CPB enhanced the functional activation of IL-17R in the hippocampus, thereby exacerbating the injury.

### 3.4. IL-17A Activated the Caspase-12-Dependent ERS Apoptosis Pathway in Hippocampus After CPB

To delineate the apoptotic pathways activated in the hippocampal region following CPB, we assessed key apoptotic markers. Post-CPB, significant increases in the activation of Caspase-8 (Figure 4a,b) and Caspase-12 (Figure 4a,d) were observed, indicative of extrinsic and ERS-mediated apoptotic pathways, respectively. Notably, IL-17A-Neutralizing mAb treatment significantly attenuated Caspase-12 activation but had no effect on Caspase-8 levels, suggesting that IL-17A exacerbates hippocampal apoptosis specifically through the ERS pathway rather than the extrinsic pathway. In contrast, Caspase-9 (Figure 4a,c), a marker of the intrinsic apoptotic pathway, showed no significant changes post-CPB or following IL-17A-Neutralizing mAb intervention, indicating minimal involvement of this pathway. Additionally, the pro-apoptotic factor Bax (Figure 4a,e) and the anti-apoptotic factor Bcl-2 (Figure 4a,f), which are involved in multiple apoptotic pathways including ERS, exhibited trends consistent with Caspase-12 activation. In summary, CPB-induced hippocampal apoptosis is predominantly mediated by IL-17A through a Caspase-12-dependent ER stress pathway, rather than other apoptotic mechanisms.

### 3.5. IL-17A Enhances CPB-Induced Neuronal Apoptosis in Hippocampus Through Act1-IRE1-JNK1 Pathway

The ERS-mediated apoptotic pathway is activated via three effectors: IRE1, PERK, and ATF6. To identify the specific ERS pathway, we analyzed key markers. Both GRP78 (Figure 5a,b) and CHOP (Figure 5a,c), indicators of overall ERS levels, were significantly upregulated post-CPB, with reductions observed following IL-17A-Neutralizing mAb or 4-PBA (an ER stress inhibitor) treatment, confirming IL-17A-driven ERS activation in the hippocampal region.

Phosphorylation levels of IRE1 (Figure 5a,d) and PERK (Figure 5a,e) and cleavage level of ATF-6 (Figure 5a,f) were elevated post-CPB. However, only IRE1 phosphorylation was markedly suppressed by IL-17A-Neutralizing mAb, indicating that IL-17A preferentially activates the IRE1 branch of ERS. JNK1 (Figure 5a,g), a downstream effector of IRE1, mirrored IRE1 activation trends.

To delineate upstream signaling, we focused on Act1—a critical adaptor for IL-17A signaling—and Hsp90, a chaperone regulating both Act1 and IRE1. Co-IP (Figure 5h) revealed enhanced Hsp90-Act1 binding post-CPB, which potentiated Act1 activity, while reduced Hsp90-IRE1 interaction facilitated IRE1 kinase domain autophosphorylation. IL-17A-Neutralizing mAb reversed these binding patterns, decreasing Hsp90-Act1 association and increasing Hsp90-IRE1 interaction.

Collectively, these data demonstrate that CPB exacerbates hippocampal neuronal apoptosis via IL-17A-dependent activation of the Act1-IRE1-JNK1 signaling axis, with Hsp90 serving as a regulatory node coordinating upstream Act1 and downstream IRE1 activities.

### 3.6. IL-17A Enhances OGD/R-Induced Apoptosis Through Act1-IRE1-JNK1 Pathway

To validate our findings at the cellular level, we employed a rat neuroblastoma (B104) cell line subjected to OGD/R. In OGD/R-treated cells, the activation levels of Caspase-3 (Figure 6a,b) and Caspase-12 (Figure 6a,c), as well as the phosphorylation levels of IRE1 (Figure 6a,d) and JNK1 (Figure 6a,e), were significantly elevated. Exogenous rmIL-17A further exacerbated these increases, while treatment with the IRE1 inhibitor GSK2850163 markedly suppressed these effects, confirming IL-17A’s role in enhancing IRE1-mediated ERS apoptosis under OGD/R conditions.

TUNEL staining (Figure 6f,g) corroborated these results: OGD/R increased apoptotic cell counts compared to normoxia controls, and this effect was further amplified by rmIL-17A. However, GSK2850163 treatment significantly reduced apoptosis, demonstrating the critical role of IRE1 in this pathway.

Co-IP assays (Figure 6h) revealed that OGD/R enhanced Hsp90-Act1 binding while reducing Hsp90-IRE1 interaction. GSK2850163 reversed these binding patterns, decreasing Hsp90-Act1 association and increasing Hsp90-IRE1 binding, consistent with our in vivo observations.

## 4. Discussion

The present study provides compelling evidence that IL-17A exacerbates CPB-induced hippocampal neuronal apoptosis through a novel Act1-IRE1-JNK1 signaling axis, with Hsp90 serving as a critical regulatory node bridging upstream inflammatory signaling and downstream ERS-mediated apoptosis. It is well known that the hippocampus is the most vulnerable brain region, particularly to ischemia–hypoxia. Similarly, our findings not only identify the hippocampus as a vulnerable brain region in CPB-related neurological injury but also delineate a previously unrecognized molecular cascade linking systemic inflammation to localized neuronal damage. These insights advance our understanding of CPB-associated neuropathology and highlight IL-17A as a promising therapeutic target for perioperative neuroprotection.

The hippocampus emerged as the primary locus of IL-17A-driven injury in our CPB model, exhibiting both the highest IL-17A elevation and most pronounced apoptotic activation. This regional specificity aligns with clinical observations of hippocampal atrophy in post-CPB cognitive dysfunction [24], yet mechanistic explanations have remained elusive. The selective vulnerability of the hippocampus to CPB-induced injury aligns with its unique neuroanatomical and metabolic characteristics. As a region with high oxygen demand and glutamatergic excitatory activity, the hippocampus is particularly susceptible to ischemia–reperfusion (I/R) injury and inflammatory insults [25]. Our observation of pronounced IL-17A elevation in the hippocampus compared to other brain regions suggests localized amplification of inflammatory signaling, potentially mediated by blood–brain barrier (BBB) disruption during CPB. Previous studies have demonstrated that IL-17A can directly increase BBB permeability through MMP-9 activation and tight junction protein degradation [26], creating a feedforward loop that facilitates cytokine infiltration into vulnerable brain regions. The hippocampus contains dense IL-17RA expression in neuronal populations [27], which may explain its heightened sensitivity. The membrane translocation assays revealed CPB-induced IL-17RA activation, suggesting that systemic IL-17A elevation directly engages hippocampal neuronal receptors.

Consistent with the conclusions of previous studies, this study found a significant increase in IL-17A after CPB [28]. IL-17A, a pro-inflammatory cytokine, has been implicated in a variety of inflammatory and autoimmune diseases. The role of IL-17A in CPB-induced brain injury represents a critical yet underexplored area in the pathophysiology of post-surgical neurological complications. Our findings demonstrate that IL-17A exacerbates hippocampal neuronal apoptosis through the activation of the Act1-IRE1-JNK1 signaling axis, highlighting its central role in mediating ERS-dependent cell death. In addition to the mechanisms elucidated in this study, prior research suggests alternative pathways through which IL-17A may exacerbate post-CPB injury. First, elevated IL-17A levels have been shown to upregulate pro-inflammatory cytokines such as IL-1, IL-6, TNF-α, and IL-8 [29], thereby amplifying the systemic inflammatory response following CPB [30,31]. Second, IL-17A plays a critical role in the recruitment, mobilization, and activation of immune cells, particularly neutrophils [29], which constitute a major inflammatory cell population during cardiopulmonary bypass [32]. These findings highlight the multifaceted role of IL-17A in driving both local and systemic inflammation, further underscoring its potential as a therapeutic target in CPB-related complications.

Our data delineate ERS-mediated apoptosis as the principal mechanism of IL-17A-induced hippocampal damage. While caspase-8 (extrinsic pathway) showed modest activation post-CPB, its insensitivity to IL-17A-Neutralizing mAb suggests that this represents a parallel inflammatory cascade rather than IL-17A-dependent signaling. In contrast, caspase-12 activation and CHOP upregulation were tightly coupled to IL-17A levels, implicating ERS as the dominant IL-17A effector. ERS is a cellular response to the accumulation of misfolded proteins in the ER lumen, leading to the activation of the unfolded protein response (UPR) [7]. While the UPR initially aims to restore cellular homeostasis, prolonged or severe ERS can trigger apoptosis [8]. Our study identifies the ERS pathway as a critical mediator of CPB-induced neuronal apoptosis, with IL-17A acting as a key upstream regulator. The current findings demonstrate that IL-17A preferentially activates the IRE1-JNK1 axis rather than PERK or ATF6 pathways, suggesting a selective mechanism for coupling cytokine signaling to apoptotic machinery.

The central role of Hsp90 in this pathway warrants particular attention. While its function in IL-17R signaling through Act1 stabilization has been described [21], our discovery of its concurrent interaction with IRE1 expands its regulatory repertoire. The competitive binding dynamics observed—where IL-17A stimulation enhances Hsp90-Act1 association while reducing Hsp90-IRE1 interaction—provide a plausible explanation for IRE1 hyperactivation. This “chaperone switching” mechanism represents a novel regulatory paradigm, as previous studies primarily focused on Hsp90’s role in stabilizing individual client proteins rather than coordinating cross-pathway interactions [33]. The role of Hsp90 in regulating IL-17A signaling is further supported by the observed effects of Hsp90 inhibitors, which have been shown to attenuate IL-17A-induced inflammation and apoptosis in various models [21,34,35]. In our study, the administration of an IL-17A-Neutralizing mAb reversed the binding patterns of Hsp90, decreasing its association with Act1 and increasing its interaction with IRE1. The dual role of Hsp90 in regulating both Act1 and IRE1 highlights its potential as a therapeutic target for mitigating CPB-induced brain injury.

The functional coupling between Act1 and IRE1 establishes a novel inflammatory–ERS crosstalk pathway. Act1 typically mediates IL-17A signaling through TRAF6-dependent NF-κB activation [36], but our findings reveal an alternative route via IRE1-JNK1. This aligns with emerging evidence that JNK1 activation during ERS can occur independently of classical UPR sensors [12,37]. Mechanistically, IL-17A may enhance IRE1 activity through two complementary mechanisms: 1) Hsp90 redistribution (as above), and 2) ROS generation secondary to Act1-TRAF6 signaling, which oxidizes IRE1’s kinase domain to prolong activation [38,39,40]. The spatial coordination of these events—with Hsp90 serving as a scaffold—likely explains the pathway’s rapid activation kinetics. Importantly, IRE1 inhibition not only blocked apoptosis but also normalized Hsp90 binding patterns, suggesting a feedback loop where IRE1 activity reinforces Act1 signaling.

The neuroprotective efficacy of IL-17A-Neutralizing mAb underscores its therapeutic potential. Clinically, IL-17A inhibitors (e.g., Secukinumab) are already approved for autoimmune disorders [41], suggesting rapid repurposing possibilities. However, systemic IL-17A blockade carries infection risks [42,43], necessitating targeted delivery approaches. Our identification of hippocampal IL-17RA activation supports the development of intranasal or cerebrospinal fluid-administered biologics to achieve CNS-specific targeting. Alternatively, small molecule inhibitors of Hsp90-Act1 interaction could disrupt pathological signaling while preserving physiological IL-17A functions. The differential effects on caspase-12 versus caspase-8 further suggest that ERS pathway modulation may provide greater neuroprotection with fewer off-target effects compared to broad-spectrum anti-apoptotics.

Neutralization of IL-17A only partially reversed CPB-induced injury, suggesting that IL-17A is at the center of the inflammatory network rather than the sole driver in the pathogenesis of post-CPB brain injury. CPB may simultaneously activate TNF-α/IL-1β signaling (via the TLR4/MyD88 pathway) and complement component C5a [44,45]. Previous studies have demonstrated that these inflammatory mediators can independently induce ERS-mediated apoptosis, independent of IL-17A. Therefore, future studies should further investigate the synergistic effects between IL-17A and other inflammatory factors, such as TNF-α, in the mechanism of brain injury following CPB.

While our study clarifies key mechanisms, several questions remain. First, while B104 cells provided a tractable model for initial pathway dissection, future studies should confirm these findings in primary neuron–glia co-cultures to fully account for the contributions of non-neuronal cells to IL-17A-mediated injury. But it is reasonable to select the B104 cell line to construct the OGD model in this study, as B104 cells retain neuronal characteristics (e.g., NMDARs, neurofilaments) and are widely used in ischemia–reperfusion research [46,47]. In addition, OGD/R (1 h/24 h) can simulate the core pathology of metabolic stress and reperfusion injury of CPB. Second, the in vivo effects of Hsp90 inhibitors warrant investigation, given their dual impact on Act1 and IRE1. Third, sex-specific differences were not addressed, as only male rats were studied. Future research on female models is needed. Fourth, the central versus peripheral origin of IL-17A in this model was not experimentally determined. Peripheral IL-17A may arise from systemic inflammation induced by CPB, prompting release by neutrophils/Th17 cells in the periphery, with subsequent infiltration into the brain through a compromised blood–brain barrier. Alternatively, CNS-derived IL-17A could originate from astrocytes, supported by prior observations of substantial IL-17A/astrocyte co-localization in murine ischemic stroke models [48]. Future studies should specifically investigate the cellular sources of IL-17A in CPB-associated neuroinflammation. Finally, long-term cognitive outcomes beyond 24 h post-CPB should be evaluated to determine whether acute IL-17A blockade confers lasting protection.

## 5. Conclusions

In conclusion, this study demonstrates that IL-17A plays a central role in exacerbating CPB-induced brain injury through the activation of the Act1-IRE1-JNK1 signaling pathway (Figure 7). The findings highlight the importance of IL-17A and ERS in mediating neuronal apoptosis in the hippocampus and identify Hsp90 as a key regulatory node in this process. These results provide a strong rationale for the development of IL-17A-Neutralizing mAb therapies and Hsp90 inhibitors as potential neuroprotective agents in patients undergoing cardiac surgery with CPB.

## Figures and Tables

**Figure 1 biomolecules-15-01134-f001:**
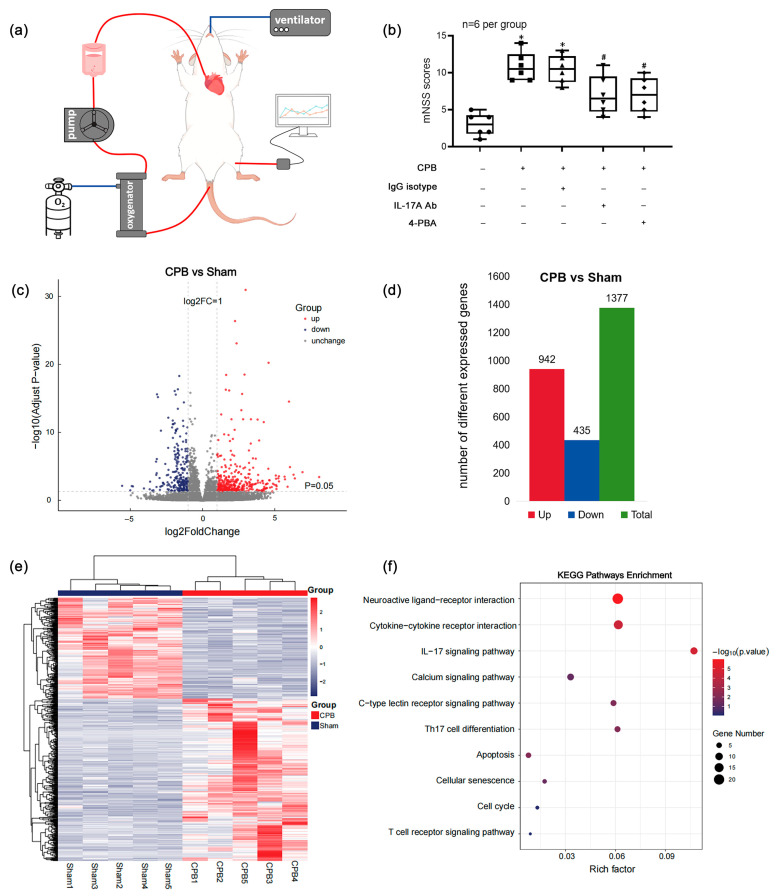
The DEGs were significantly enriched in IL-17 signaling and apoptosis-related pathways; IL-17A-Neutralizing mAb and 4-PBA can significantly improve the neurological dysfunction in rats after CPB. (**a**) A diagram of the establishment of CPB in rats. (**b**) Both IL-17A-Neutralizing mAb and 4-PBA could significantly reverse the increase in mNSS in rats after CPB. * *p* < 0.05 compared to the Sham group; # *p* < 0.05 compared to the CPB group, *n* = 6 per group. (**c**) Volcano plots of DEGs in the CPB and Sham groups. (**d**) Identification of DEGs. (**e**) Heatmap of DEGs. (**c**,**e**) A total of 1377 DEGs were screened in the CPB group including 942 upregulated and 435 downregulated compared to the Sham group. (**f**) KEGG enrichment analysis showed that DEGs in the CPB group were significantly enriched in several pathways, including the IL-17 signaling pathway and apoptosis-related pathway. *p* < 0.05, *n* = 5 per group.

**Figure 2 biomolecules-15-01134-f002:**
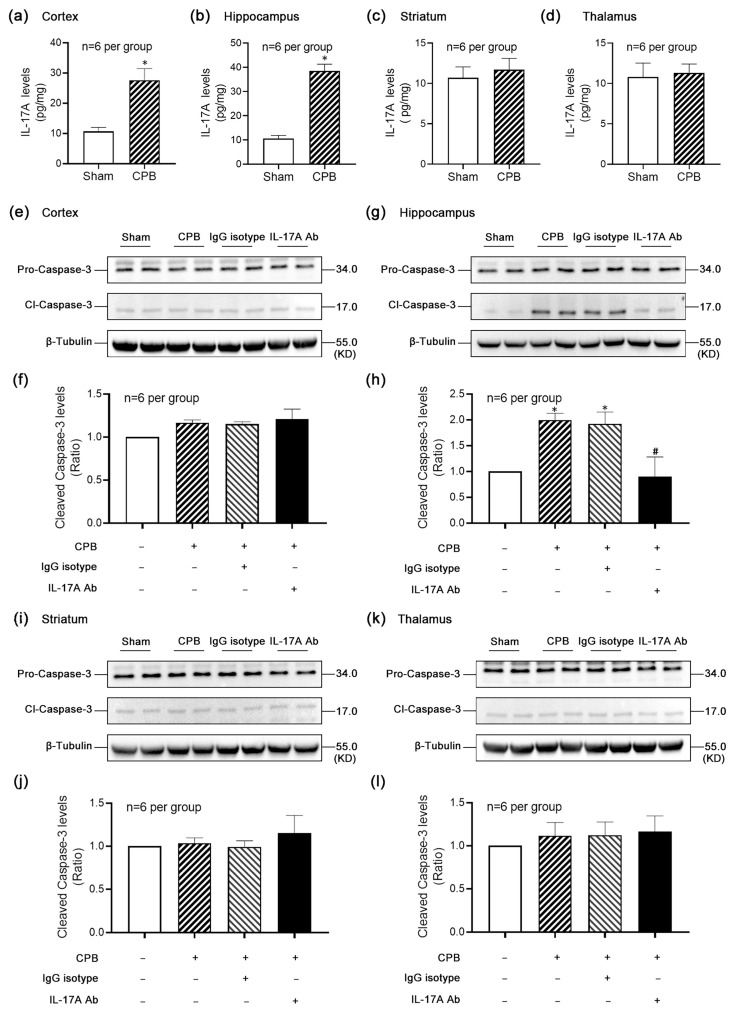
The levels of IL-17A and neuronal apoptosis in the hippocampus were significantly increased after CPB. (**a**–**d**) IL-17A levels in different brain regions of rats after CPB were detected by ELISA. The results showed that the levels of IL-17A in the cortex (**a**) and hippocampus (**b**) were significantly increased after CPB, while the levels of IL-17A in the striatum (**c**) and thalamus (**d**) were not significantly different after CPB. Typical Western blot results showed that among the four brain regions of the cortex (**e**,**f**), hippocampus (**g**,**h**), striatum (**i**,**j**), and thalamus (**k**,**l**), only the hippocampus (**g**,**h**) showed a significant increase in Cleaved-Caspase-3 level in brain tissue after CPB, representing a significant increase in apoptosis level; IL-17A-Neutralizing mAb could significantly reverse this elevation. There was no significant difference between the IgG isotype group and the CPB group, which excluded the influence of the antibody itself. * *p* < 0.05 compared to the Sham group; # *p* < 0.05 compared to the CPB group, *n* = 6 per group.

**Figure 3 biomolecules-15-01134-f003:**
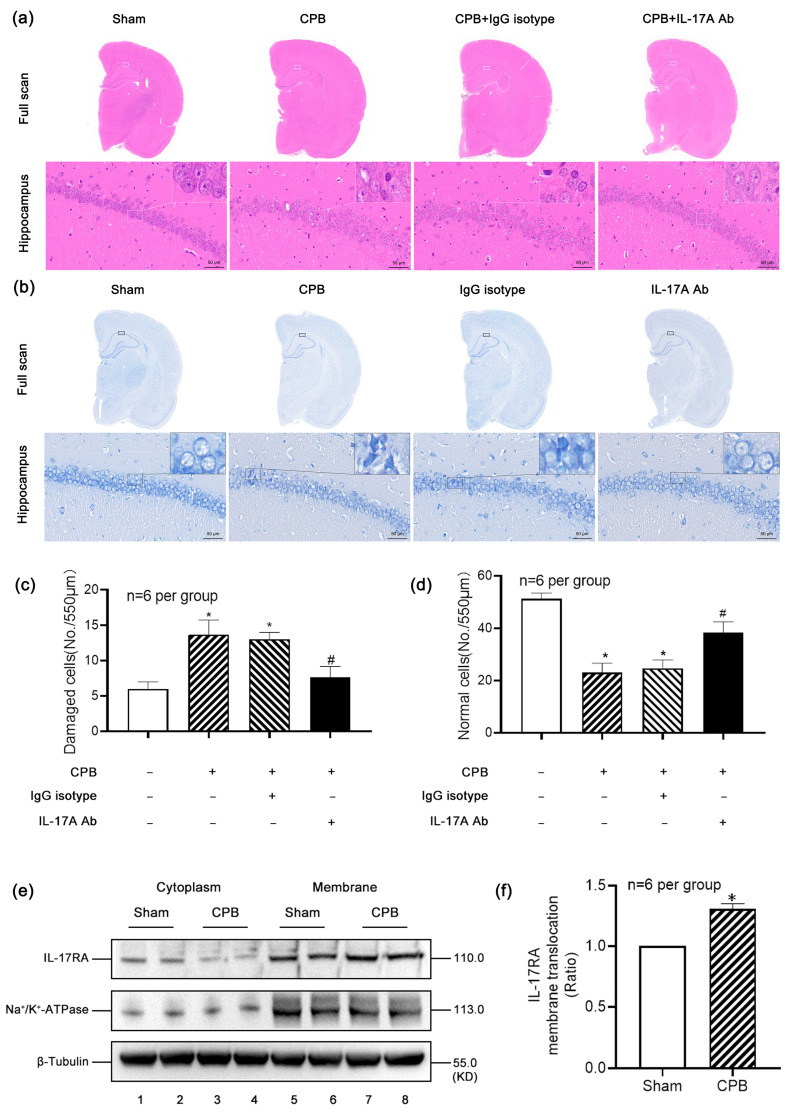
IL-17A aggravated the CPB-induced hippocampal injury, which could be significantly reversed by IL-17A-Neutralizing mAb. (**a**,**c**) Hematoxylin–eosin (H&E) staining was performed on the hippocampi of rats, and the neurons in CA1 region were observed. It was found that compared with the Sham group, the hippocampal neurons in the CPB group and the IgG isotype group were loose and irregular, and there were more damaged cells (the shape of the damaged cells was irregular, the cytoplasm was stained deeper, and the nuclear structure was difficult to identify). After the intervention of IL-17A-Neutralizing mAb, the neuronal arrangement in the CA1 region of the rat hippocampus recovered in an orderly manner and the number of damaged cells decreased. (**b**,**d**) The results of Nissl staining were consistent with those of H&E staining, and the cells that could clearly identify the nuclear and cytoplasmic structures were identified as normal nerve cells. The number of normal nerve cells in the CPB and IgG isotype groups decreased compared with the Sham group. After the intervention of IL-17A-Neutralizing mAb, the number of normal nerve cells recovered somewhat. (**e**,**f**) Typical Western blot results showed that the membrane translocation level of IL-17RA was increased after CPB. * *p* < 0.05 compared to the Sham group; # *p* < 0.05 compared to the CPB group, *n* = 6 per group.

**Figure 4 biomolecules-15-01134-f004:**
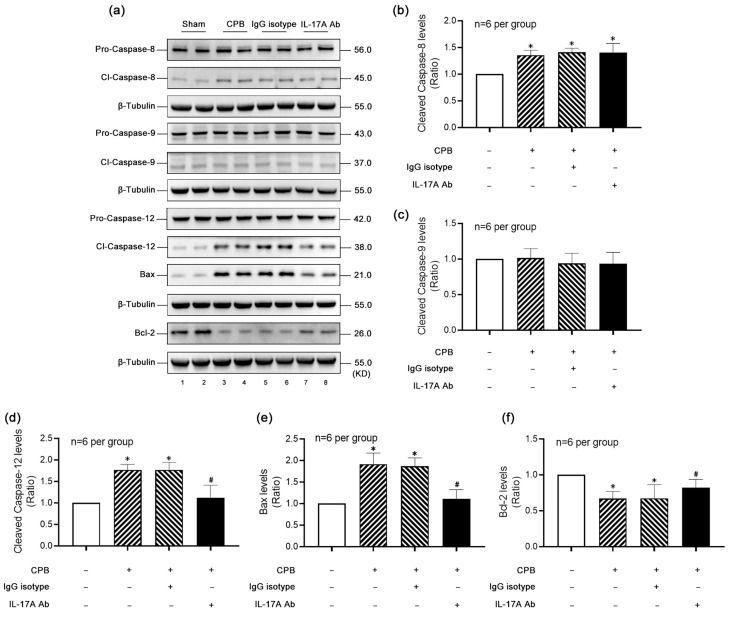
IL-17A activated the Caspase-12-dependent ERS apoptosis pathway in hippocampus after CPB. (**a**) Western blot detection of typical proteins on different apoptosis pathways showed that (**b**) compared with the Sham group, Cleaved-Caspase-8 levels were significantly increased in the CPB and the IgG isotype groups, but IL-17A-Neutralizing mAb could not inhibit the increase; (**c**) Cleaved-Caspase-9 levels were not significantly different among groups; compared with the Sham group, Cleaved-Caspase-12 levels (**d**) and Bax levels (**e**) in the CPB group and the IgG isotype group were significantly increased, and Bcl-2 levels (**f**) were significantly decreased; intervention with IL-17A-Neutralizing mAb significantly inhibited the above trends. * *p* < 0.05 compared to the Sham group; # *p* < 0.05 compared to the CPB group, *n* = 6 per group.

**Figure 5 biomolecules-15-01134-f005:**
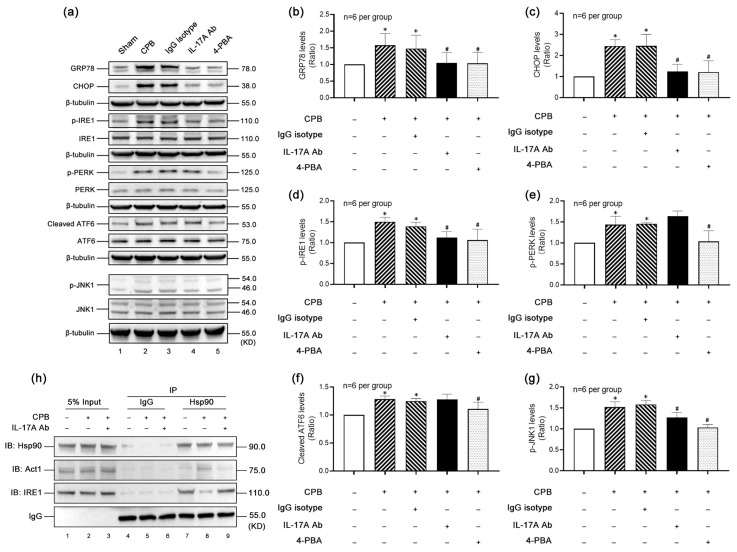
IL-17A enhances CPB-induced neuronal apoptosis in the hippocampus through the Act1-IRE1-JNK1 pathway. (**a**) Typical Western blot results showed that the expression of GRP78 (**b**) and CHOP (**c**) increased in the hippocampal brain tissue of rats after CPB operation, and the increase was significantly inhibited after the intervention of IL-17A-Neutralizing mAb and 4-PBA. Therefore, CPB promotes the activation of ERS in the hippocampus mediated by IL-17A. The phosphorylation levels of and cleavage levels of three ERS activating effectors, IRE1 (**d**), PERK (**e**), and ATF-6 (**f**), all increased significantly after CPB, but only the phosphorylation level of IRE1 (**d**) was significantly inhibited by IL-17A-Neutralizing mAb. Therefore, CPB induced IL-17A to activate IRE1-mediated ERS. JNK1 (**g**), as a key signaling molecule downstream of IRE1, has the same change trend as IRE1. (**h**) At the animal level, after CPB, the binding of Hsp90 to Act1 increased, but the binding of HSP90 to IRE1 decreased. After the intervention of IL-17A-Neutralizing mAb, the binding of Hsp90 to Act1 decreased, while the binding of HSP90 to IRE1 increased. * *p* < 0.05 compared to the Sham group; # *p* < 0.05 compared to the CPB group, *n* = 6 per group.

**Figure 6 biomolecules-15-01134-f006:**
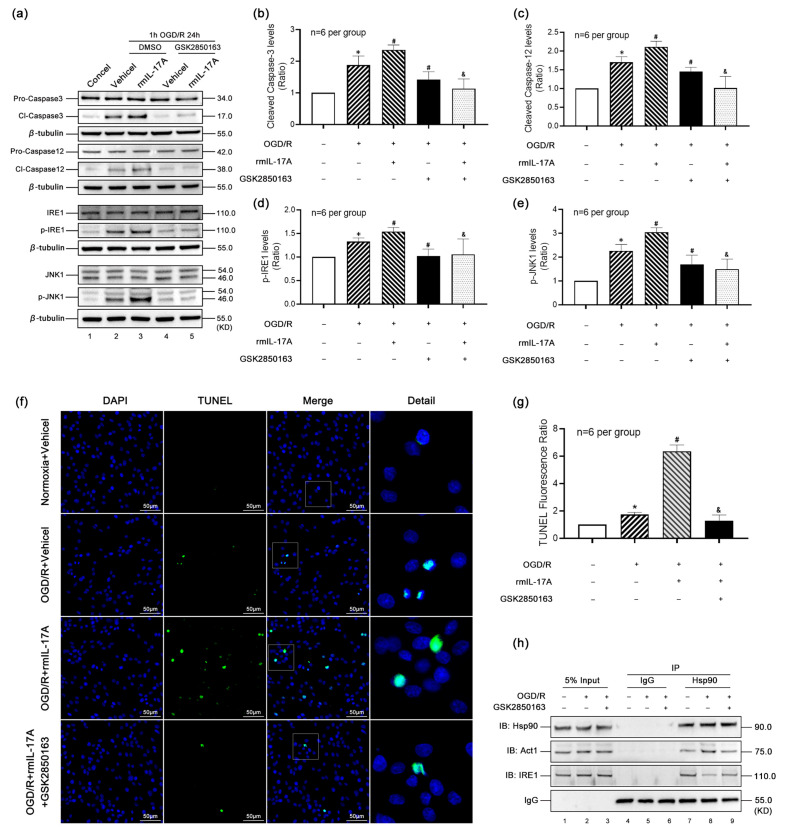
IL-17A enhances OGD/R-induced apoptosis through the Act1-IRE1-JNK1 pathway. (**a**–**e**) Apoptosis- and ERS-related protein expression levels in different B104 cell groups. (**a**) Typical Western blot bands showed that compared with the Concel group, the hydrolysis levels of Caspase-3 (**b**) and Caspase-12 (**c**) in B104 cells treated with OGD/R were significantly increased, and the phosphorylation levels of IRE1 (**d**) and JNK1 (**e**) were also significantly increased. On this basis, the hydrolysis levels of Caspase-3 and -12 and the phosphorylation levels of IRE1 and JNK1 were further increased after the intervention of rmIL-17A. After intervention with GSK2850163, the above rising trend was significantly suppressed. * *p* < 0.05 compared to the Concel group; # *p* < 0.05 compared to the OGD/R++DMSO+ Vehicel group; & *p* < 0.05 compared to the OGD/R+DMSA+rmIL-17A group, *n* = 6 per group; (**f**,**g**) TUNEL staining results of B104 cell in different groups: compared with the Normoxia group, the number of apoptotic cells in B104 cell treated with OGD/R was significantly increased, and rmIL-17A could significantly increase the apoptosis induced by OGD/R. After the intervention of GSK2850163, the increase in apoptosis induced by rmIL-17A and OGD/R was significantly reversed. * *p* < 0.05 compared to the Normoxia+Vehicel group; # *p* < 0.05 compared to the OGD/R+Vehicel group; & *p* < 0.05 compared to the OGD/R+rmIL-17A group, *n* = 6 per group. (**h**) At the cellular level, after OGD/R treatment, the binding of Hsp90 to Act1 increased, but the binding of HSP90 to IRE1 decreased. After intervention with GSK2850163, the binding of Hsp90 to Act1 decreased, while the binding of HSP90 to IRE1 increased.

**Figure 7 biomolecules-15-01134-f007:**
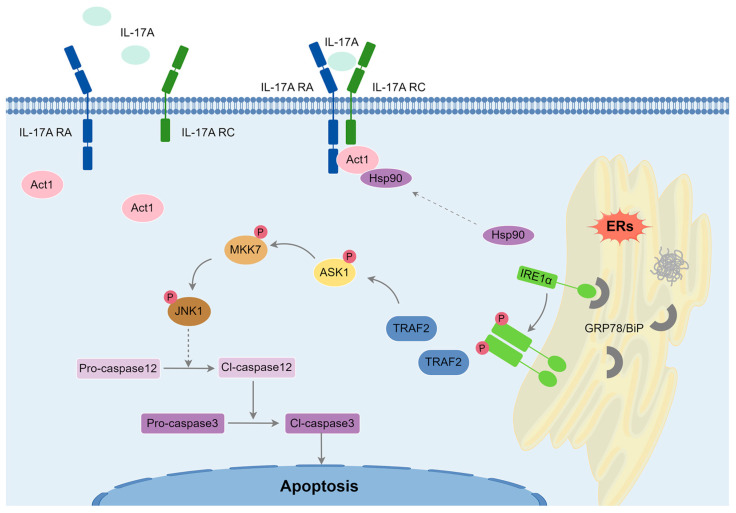
The schematic diagram shows that IL-17A aggravates Caspase-12-dependent neuronal apoptosis after CPB through the Act1-IRE1-JNK1 signaling pathway. IL-17A acts on the receptor complex formed by IL-17RA and IL-17RC to recruit Act1 and increase the binding of Act1 to Hsp90, promoting Act1 signal transmission. It also reduces the binding of Hsp90 to IRE1, promoting IRE1 phosphorylation. After the phosphorylation of IRE1, a dimer is formed, and the signal is transmitted downward to activate JNK1, which ultimately promotes the activation of Caspase-12 and increases apoptosis.

## Data Availability

The data that support the findings of this study are available from the Appendix A/corresponding authors upon reasonable request.

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
