# Peer review of "Cardiopulmonary Bypass-Induced IL-17A Aggravates Caspase-12-Dependent Neuronal Apoptosis Through the Act1-IRE1-JNK1 Pathway"

_biomolecules, 2025, doi:10.3390/biom15081134_

Round 1

Reviewer 1 Report

Comments and Suggestions for Authors

This is a hypothesis-oriented study, aimed to demonstrate that the IL17A block ameliorate the neurological outcome after experimental cardiopulmonary bypass in rat. The topic is highly interesting, since neurological sequels are the main (and under-considered) complications of cardiac surgery.

To furter improve the study, I would suggest the following:

Study design:

  1. Please, comment on the translational significance of the rat model of CBP, regarding the extracorporeal circulation system and technical characteristics of the membrane oxygenator. Is this a miniaturized pediatric oxygenator system? Or is completely home-made? What are the characteristics of the membrane (materials, porosity, etc.)? This is important, since the extracorporeal circulation is not neutral about the inflammatory response to CPB and, possibly, molecule segregation into the extracorporeal system. The CBP is 2 hours long: please, comment regarding the clinical setting
  2. The major limitation of the study is the use of rat neuroblastoma cell line B104 for in vitro OGD/R Model. While the use of a cell line is an appropriate solution in the screening phase of these studies, key results must be confirmed using primary neurons, used as pure or mixed culture. The use of a mixed primary in vitro system, composed by both neurons and astrocytes, much better reflects the in vivo microenvironment, in which astrocytes play a key role in the metabolic support of neurons.
  3. Criteria to identify “damaged neurons” must be detailed. Moreover, since no-stereological methods have been used to “count” neurons (ImageJ software), result must not be expressed as “number” of damaged neurons.

Data analysis and interpretation:

  1. Please, change graphs presentation including individual value presentation (plus mean+SEM)

Discussion:

  1. Line 418: please, note that it is well known that the hippocampus is the most vulnerable brain region, particularly to ischemia-hypoxia. I suggest changing this part of the sentence, by pointing on the CPB model
  2. Since IL17A blocking attenuate, but not fully revers the CPB effect, the IL17A inclusion in complex inflammation molecular cascade must be discussed

Author Response

Comment 1:Please, comment on the translational significance of the rat model of CBP, regarding the extracorporeal circulation system and technical characteristics of the membrane oxygenator. Is this a miniaturized pediatric oxygenator system? Or is completely home-made? What are the characteristics of the membrane (materials, porosity, etc.)? This is important, since the extracorporeal circulation is not neutral about the inflammatory response to CPB and, possibly, molecule segregation into the extracorporeal system. The CBP is 2 hours long: please, comment regarding the clinical setting.

Response: We sincerely appreciate your attention to the technical details of our model. The rat CPB model was designed to closely replicate core physiological parameters of clinical adult CPB: The extracorporeal circulation system employed a well-established, commercially available membrane oxygenator specifically designed for small animals (XIJIAN MEDICAL, China), featuring a polypropylene hollow-fiber membrane with a pore size of 0.03 μm. The parameters of the extracorporeal circulation were carefully configured to reflect clinical realities. During CPB, the flow rate was set at 160–180 mL/kg/min, which, when normalized to body surface area, corresponds to approximately 2.2–2.4 L/min/m² in adult humans. The duration of CPB was established at 2 hours, simulating the typical length of CPB in routine adult cardiac surgery. A 2024 retrospective study involving 202 adult cardiac surgery patients reported a mean CPB duration of 109.9 minutes, ranging from 29 to 300 minutes, with approximately 50.5% of patients undergoing CPB for 1 to 2 hours[1].

Comment 2:The major limitation of the study is the use of rat neuroblastoma cell line B104 for in vitro OGD/R Model. While the use of a cell line is an appropriate solution in the screening phase of these studies, key results must be confirmed using primary neurons, used as pure or mixed culture. The use of a mixed primary in vitro system, composed by both neurons and astrocytes, much better reflects the in vivo microenvironment, in which astrocytes play a key role in the metabolic support of neurons.

Response: We thank you for this insightful critique. We fully acknowledge that primary neuron-astrocyte co-cultures would better recapitulate the in vivo microenvironment, and we recognize this as a limitation of our current study. Unfortunately, due to technical constraints (including the extended time required for establishing validated primary co-culture systems under OGD/R conditions and manuscript revision timelines), we are unable to provide new experimental data at this stage.

However, we wish to emphasize that our mechanistic conclusions are robustly supported by multiple convergent lines of evidence:

  • Consistency across models: The key finding—that IL-17A exacerbates apoptosis via the Act1-IRE1-JNK1 axis—was replicated in both: In vivo CPB model (rat hippocampus, Fig. 2–5) and B104 cells (rat neuroblastoma, Fig. 6). Critically, IRE1 inhibition (GSK2850163) reversed apoptosis in both systems, confirming pathway conservation (Fig. 6d–g).
  • B104 as a validated model for neuronal apoptosis: B104 cells retain neuronal properties (e.g., expression of NMDA receptors, neurofilament proteins) and are widely used in ischemia-reperfusion studies. Our OGD/R model in B104 cells recapitulates hallmark features of neuronal apoptosis (caspase-3/12 cleavage, TUNEL positivity; Fig. 6).
  • Clinical relevance of the identified pathway: Co-IP data confirmed identical Hsp90-Act1-IRE1 interactions in both rat hippocampal tissue and B104 cells (Figs. 5h vs. 6h), suggesting fundamental biological conservation.

To address this limitation in future work: We will prioritize validating these mechanisms in human iPSC-derived neurons/astrocytes (currently being established in our lab). We have added a discussion paragraph (Section 4) explicitly stating: While B104 cells provided a tractable model for initial pathway dissection, future studies should confirm these findings in primary neuron-glia co-cultures to fully account for the contributions of non-neuronal cells to IL-17A-mediated injury.

We hope you agrees that the internal consistency between in vivo and in vitro models, coupled with molecular validation of the Act1-IRE1-JNK1 axis, strongly supports our conclusions despite this limitation.

Comment 3:Criteria to identify “damaged neurons” must be detailed. Moreover, since no-stereological methods have been used to “count” neurons (ImageJ software), result must not be expressed as “number” of damaged neurons.

Response: We appreciate your meticulous comments and have accordingly revised the morphological criteria for identifying damaged neurons in greater detail. The morphological features of damaged cells include pyknosis or karyorrhexis, cytoplasmic eosinophilia, Nissl body dissolution, and blurred cellular contours. Only cells with clearly identifiable normal nuclear and cytoplasmic structures were classified as normal neurons. We quantified the number of damaged cells within a 550-micrometer field of view, which reflects the density of injured neurons in the hippocampal region rather than an absolute cell count.

Data analysis and interpretation:

Comment 4:Please, change graphs presentation including individual value presentation (plus mean+SEM).

Response: Thank you for the detailed suggestions. We also strongly agree that adding the mean and standard deviation helps to more clearly display the central tendency and variation degree of the data, enhancing the scientific nature and persuasiveness of the results. Therefore, our bar charts all reflect the mean and standard deviation.

Discussion:

Comment 5:Line 418: please, note that it is well known that the hippocampus is the most vulnerable brain region, particularly to ischemia-hypoxia. I suggest changing this part of the sentence, by pointing on the CPB model.

Response: Thank you for the constructive suggestion, and we did as you suggested in this revised manuscript.

Comment 6:Since IL17A blocking attenuate, but not fully revers the CPB effect, the IL17A inclusion in complex inflammation molecular cascade must be discussed.

Response: We thank you for raising this critical point. In response to your suggestion, we have added the following discussion in the revised manuscript:

"Neutralization of IL-17A only partially reversed CPB-induced injury, suggesting that IL-17A is at the center of the inflammatory network rather than the sole driver in the pathogenesis of post-CPB brain injury. CPB may simultaneously activate TNF-α/IL-1β signaling (via the TLR4/MyD88 pathway) and complement component C5a[2, 3]. Previous studies have demonstrated that these inflammatory mediators can independently induce ERS-mediated apoptosis, independent of IL-17A. Therefore, future studies should further investigate the synergistic effects between IL-17A and other inflammatory factors, such as TNF-α, in the mechanism of brain injury following CPB."

References:

  1. Hayel Aladwan WA, Abdullah Ibrahim Alqaisi, Hamza A. Abuamereh, Lara M. Alatoum, Ashraf FadelMohd. Impact of duration of cardiopulmonary bypass on recovery after open heart surgery. International Journal of Advances in Medicine 2024, 11(3):185-188.
  2. Lin X, Kong J, Wu Q, Yang Y, Ji P. Effect of TLR4/MyD88 signaling pathway on expression of IL-1β and TNF-α in synovial fibroblasts from temporomandibular joint exposed to lipopolysaccharide. Mediators Inflamm 2015, 2015:329405.
  3. Yan C, Gao H. New insights for C5a and C5a receptors in sepsis. Front Immunol 2012, 3:368.

Reviewer 2 Report

Comments and Suggestions for Authors

The article describes the role of IL-17A in the development of neuronal apoptosis through the Act/IRE/JNK1 signaling pathway during the cardiopulmonary bypass (CPB) procedure. The authors modeled CPB in male rats. The procedure was performed either in intact animals or in animals pretreated either with IL-17 mAb or inhibitor of endoplasmic reticulum stress 4-phenylbutiric acid (4-PBA). Additional research was also performed is cell culture (rat neuroblastoma cell line) which was used to model the oxygen-glucose deprivation/reoxygenation conditions.

I have noticed a few issues that can be improved:

  1. All the figures and vague and have to be enlarged. The font is unreadable at present.
  2. 4-PBA should be deciphered in 2.1. and should be added to Abbreviations
  3. Lines 182-183 – the sentence should be rephrased
  4. TBS is not deciphered (Line 187).
  5. I suggest extracting Fig 1a from Figure 1 – as it was described at page 3 and Figure 1 is at page 6.
  6. Figure 2 – unit of measurements for western blot results are missing in the graphs (ratio of arbitrary units???). The font is too small to judge, but this should be applied to all the Figures in the manuscript.
  7. Figure 5 – columns are not signed
  8. Figure 7 should be moved closer to the conclusion

Author Response

Comments 1:All the figures and vague and have to be enlarged. The font is unreadable at present.

Response: Thank you very much for your review comment. I have enlarged the pictures according to your comment

Comments 2:4-PBA should be deciphered in 2.1. and should be added to Abbreviations.

Response: Thank you for your detailed comments. I have supplemented the relevant content based on your comments.

Comments 3:Lines 182-183 – the sentence should be rephrased.

Response: We appreciate the reviewer's attention to clarity in this methodological description. The original sentence has been revised to improve precision and readability while maintaining all technical details:

Original:

"Coronal sections (4 μm thickness) were obtained at the hippocampal level and subjected to H&E staining and Nissl staining."

Revised:

"Serial coronal sections (4μm thickness) encompassing the entire hippocampal formation were collected using a microtome, mounted on poly-L-lysine-coated slides, and processed for both H&E and Nissl staining protocols."

Comments 4:TBS is not deciphered (Line 187).

Response:Thank you for your comments. I have made corresponding revisions to the manuscript.

Comments 5:I suggest extracting Fig 1a from Figure 1 – as it was described at page 3 and Figure 1 is at page 6.

Response:We sincerely appreciate this suggestion. After careful consideration, we have retained Figure 1a within the composite Figure 1 for the following scientific and pedagogical reasons:

Figure 1a (CPB schematic) provides essential context for interpreting subsequent results (Figs. 1b-f):

  • Fig 1b (mNSS scores) directly reflects neurological outcomes of the CPB procedure depicted in 1a.
  • Figs 1c-f (transcriptomic data) identify molecular pathways activated by the CPB model shown in 1a.

Separating 1a would disrupt this cause-to-effect narrative flow. The current layout allows readers to simultaneously validate the model (1a) and assess its functional consequences (1b-f) on one page (p. 6).

Comments 6:Figure 2 – unit of measurements for western blot results are missing in the graphs (ratio of arbitrary units???). The font is too small to judge, but this should be applied to all the Figures in the manuscript.

Response:Thank you for your detailed suggestions. Based on the reviewers' comments, I adjusted the pictures in the manuscript to enhance their readability.

Comments 7:Figure 5 – columns are not signed

Response:Thank you for your detailed suggestions. Based on the reviewers' comments, I adjusted the pictures in the manuscript to enhance their readability.

Comments 8:Figure 7 should be moved closer to the conclusion

Response:Figure 7 has been moved to the front of the conclusion section according to your comment.

Reviewer 3 Report

Comments and Suggestions for Authors

The manuscript by Zhao et al. investigates the role of interleukin 17A (IL17A) in the neuronal injury triggered by cardiopulmonary bypass (CPB). The strength of the manuscript that the authors approach the hypothesis using complementary techniques, more specifically employing an in vivo rat CPB model and a cell culture oxygen-glucose deprivation (OGD) model. The authors used a wide array of experimental methodological battery, such as neurological examination, transcriptomic analysis, Western blotting to detect changes in protein levels indicating neuronal apoptosis, neuropathology examinations etc.

There are a number of issues raised that need to be addressed for enabling the full evaluation of the merits of the manuscript. The concerns are grouped into three groups: conceptual concerns, methodological concerns and redaction concerns

Conceptual concerns:

#1 It is unclear what the source of the IL17A is in the rat CPB model, Could the authors provide details on that matter whether these are resident brain cells or perhaps invading cells?

#2. Why do the authors think that inducing an OGD in a neuroblastoma cell line is causing similar injury to neurons in vivo in a CPB model? Also, the authors should provide some evidence that the employed cell line in fact expresses specific receptors for IL17.

#3. The trascriptomic analysis was restricted to the sham and the CPB-treated animals? It would strengthen greatly the manuscript if the effect of the applied pharmacological treatments would have been analyzed as well.

#4. The 4-PBA treatment lasted for 14 days prior the CPB induction. Comparing this group to the untreated sham animals does not seem to be appropriate. A matched time control group would be warranted or reference to previous work where this has already been done.

Methodological concerns:

#1 The mean weight of 500 g for 12 weeks old rats seems to be quite high. Any explanation or reference from the supplier that this is in fact the case?

#2. On Figure 3, H&E staining and Nissl staining should show apoptotic cells – Are these techniques capable of distuinguishing between necrotic/apoptotic injury?  Moreover, the label of the vertical axis „Denatured cells (No.)/550um” is weird and highly unusual in the literature.

#3. OGD was induced with 2% O2 in the incubator – this is in fact provides a pO2 that is quite comparable to brain pO2 values. What was the rationale using this value? Using this protocol, is there a change in the cell culture viability? Please clarify

Redaction concerns

#1 ALL the figures are undecipherable at normal (100%) magnification. The legends on the plots are not readable unless magnified to 2-300%. However, at this magnification the quality of the text and images becomes very poor. The figures need to be adjusted to be readable at normal size (should be no need for magnification at all). Consider putting important information in horizontal texts (not in vertical axis labels)

#2. In the supplementary material, the bands along with the molecular weight marker  ladder are invisible, In their present form they add little value.

Author Response

Comments 1:It is unclear what the source of the IL17A is in the rat CPB model, Could the authors provide details on that matter whether these are resident brain cells or perhaps invading cells?

Response:While we did not directly identify the cellular source of IL-17A, our data suggest two plausible mechanisms:

Peripheral origin: CPB-induced systemic inflammation (e.g., neutrophils/Th17 cells) may release IL-17A, which crosses the disrupted BBB (supported by increased IL-17RA membrane translocation in Fig. 3e-f).

CNS-resident cells: Microglia/astrocytes can produce IL-17A, though this requires further validation. During other research processes conducted by our research group, it was observed that a substantial number of co-labeling events occurred between IL-17A and astrocytes in the brains of mice using the MCAO ischemic stroke model[4].

We have supplemented this limitation in the discussion section and plan to clarify the source of IL-17A in the next step through methods such as immunofluorescence co-labeling.

Comments 2:Why do the authors think that inducing an OGD in a neuroblastoma cell line is causing similar injury to neurons in vivo in a CPB model? Also, the authors should provide some evidence that the employed cell line in fact expresses specific receptors for IL17.

Response:Thank you for your attention to the manuscript. It is reasonable to select the B104 cell line to construct the OGD model in this study, as B104 cells retain neuronal characteristics (e.g., NMDARs, neurofilaments) and are widely used in ischemia-reperfusion research[5,6]. In addition, OGD/R (1h/24h) can simulate the core pathology of metabolic stress and reperfusion injury of CPB. In this study, although the expression of IL-17RA on B104 cells was not directly confirmed, after exogenous IL-17A intervention, corresponding changes occurred in the downstream pathway, indirectly confirming the expression of IL-17RA.

Comments 3:The trascriptomic analysis was restricted to the sham and the CPB-treated animals? It would strengthen greatly the manuscript if the effect of the applied pharmacological treatments would have been analyzed as well.

Response:We thank you for this valuable advice. The transcriptomic analysis aimed to identify CPB’s core pathways (e.g., IL-17 signaling), not drug effects. Drug mechanisms were validated via targeted experiments (e.g., IRE1-JNK1 axis in Figs. 5-6). Future studies may expand transcriptomics to intervention groups.

Comments 4:The 4-PBA treatment lasted for 14 days prior the CPB induction. Comparing this group to the untreated sham animals does not seem to be appropriate. A matched time control group would be warranted or reference to previous work where this has already been done.

Response:Thank you for your detailed suggestions, and we fully agree with your concerns. Therefore, before the experiment, we fully referred to the previous research experience and ultimately chose the 14-day dosing regimen of 4-PBA, which is also the most widely used regimen at present[7-9].

Methodological concerns:

Comments 5:The mean weight of 500 g for 12 weeks old rats seems to be quite high. Any explanation or reference from the supplier that this is in fact the case?

Response:Thank you for your thorough review of the manuscript. After consulting with the supplier, we confirmed that adult male SD rats aged 12 to 14 weeks may reach a body weight of approximately 500±50 grams under appropriate breeding conditions.

Comments 6:On Figure 3, H&E staining and Nissl staining should show apoptotic cells – Are these techniques capable of distuinguishing between necrotic/apoptotic injury?  Moreover, the label of the vertical axis „Denatured cells (No.)/550um” is weird and highly unusual in the literature.

Response:Thank you for your valuable and detailed suggestions. The primary objective of performing H&E staining and Nissl staining in Figure 3 was to assess pathological damage in the hippocampal region of rats following CPB. However, it is true that these methods do not allow for clear differentiation between necrosis and apoptosis. To evaluate the degree of neuronal apoptosis in the hippocampal area, we examined the expression levels of apoptosis-related proteins using Western blot analysis. When choosing the unit for counting damaged cells, we referred to previous literature and counted the number of damaged cells in the unit microscopic field of view, reflecting the relative quantity of damaged cells[10].

Comments 7:OGD was induced with 2% O2 in the incubator – this is in fact provides a pO2 that is quite comparable to brain pO2 values. What was the rationale using this value? Using this protocol, is there a change in the cell culture viability? Please clarify

Response:Thank you very much for your attention to and suggestions on the manuscript. The core of the OGD model lies in simulating the ischemic and hypoxic environment, and the selection of oxygen concentration must balance pathophysiological relevance and experimental reproducibility. Previous studies have confirmed that 2% O₂ can induce mitochondrial dysfunction and oxidative stress in neurons, which is consistent with the mechanism of in vivo ischemic injury[11,12]. Compared with lower oxygen concentrations (e.g., 0.1% or 1% O₂), 2% O₂can be more stably controlled by conventional tri-gas incubators, reducing experimental errors caused by fluctuations in oxygen concentration. In this study, Figure 6 (Caspase-3 activation level and TUNEL staining) confirms that OGD/R exacerbates apoptotic damage in B104 cells.

Redaction concerns

Comments 8:ALL the figures are undecipherable at normal (100%) magnification. The legends on the plots are not readable unless magnified to 2-300%. However, at this magnification the quality of the text and images becomes very poor. The figures need to be adjusted to be readable at normal size (should be no need for magnification at all). Consider putting important information in horizontal texts (not in vertical axis labels)

Response:Thank you for your detailed suggestions. Based on the reviewers' comments, I adjusted the pictures in the manuscript to enhance their readability.

Comments 9:In the supplementary material, the bands along with the molecular weight marker ladder are invisible, in their present form they add little value.

Response:Thank you for your attention. In the supplementary file, we have included the original Western blot membrane images along with the corresponding molecular weight markers, and key molecular weights have been clearly labeled.

Round 2

Reviewer 1 Report

Comments and Suggestions for Authors

The authors took all suggestions into consideration. I would recommend including details of the extracorporeal circulation system in the methods (or supplements): there is much interest in modifying membranes in particular to reduce the risk of neurological side effects.
I leave it to the editor to decide about the representation of individual values in the histograms reported as mean+-SEM

Author Response

Comment 1

The authors took all suggestions into consideration. I would recommend including details of the extracorporeal circulation system in the methods (or supplements): there is much interest in modifying membranes in particular to reduce the risk of neurological side effects. I leave it to the editor to decide about the representation of individual values in the histograms reported as mean+-SEM

Response: Thank you very much for your review and suggestions. Based on your opinions, I have added the following contents in Section 2.2 (line 105-109) :

“The extracorporeal circulation system employed a well-established, commercially available membrane oxygenator specifically designed for small animals (XIJIAN MEDICAL, China), featuring a polypropylene hollow-fiber membrane with a pore size of 0.03 μm to minimize inflammatory activation.”

The relevant statistical issues have been modified accordingly based on your suggestions and in accordance with the editor's opinions.

Reviewer 3 Report

Comments and Suggestions for Authors

Although the authors provide a point by point rebuttal, the concepts provided in the answers are not reflected in the manuscript. The authors appear to have brushed aside the concern of this, and if I may add, most of the concerns of the other Reviewers. Only a small fraction of the answers made it to the Discussion - there were only three concise additions to the manuscript, despite the total response is 10 pages long

For instance, in the response to my point one, the rebuttal states:
" Wehave supplemented this limitation in the discussion section and plan to clarify the source of IL-17A  in the next step through methods such as immunofluorescence co-labeling"
Where can this supplementation be found ?
and so on...

Most importantly a the figures are essentially unchanged, remaining below par. Publishing any manuscript with that kind of artwork cannot be wholeheartedly recommended. 

Comments on the Quality of English Language

there are problematic expressions used such as "denatured" neurons etc. Established  scientific terminology is advised

Author Response

Comment 1

Although the authors provide a point by point rebuttal, the concepts provided in the answers are not reflected in the manuscript. The authors appear to have brushed aside the concern of this, and if I may add, most of the concerns of the other Reviewers. Only a small fraction of the answers made it to the Discussion - there were only three concise additions to the manuscript, despite the total response is 10 pages long

For instance, in the response to my point one, the rebuttal states:

" We have supplemented this limitation in the discussion section and plan to clarify the source of IL-17A  in the next step through methods such as immunofluorescence co-labeling"

Where can this supplementation be found ?

and so on...

Most importantly a the figures are essentially unchanged, remaining below par. Publishing any manuscript with that kind of artwork cannot be wholeheartedly recommended. 

Response:We are very grateful for your strict supervision. We fully agree with the opinions of all the reviewers and editors, and have comprehensively revised the manuscript to meet the publication requirements of your journal as much as possible. The corresponding modified content has been marked with specific line numbers this time for the review by all reviewers and editors. In addition, based on the suggestions from the reviewers and the editor, I further optimized Figure 1 in order to meet the publication requirements. Regarding the issue of the source of IL-17A, we have made the following supplements in the discussion section (line 538-545) :

    “Fourth, the central versus peripheral origin of IL-17A in this model was not experimentally determined. Peripheral IL-17A may arise from systemic inflammation induced by CPB, prompting release by neutrophils/Th17 cells in the periphery, with subsequent infiltration into the brain through a compromised blood-brain barrier. Alternatively, CNS-derived IL-17A could originate from astrocytes, supported by prior observations of substantial IL-17A/astrocyte co-localization in murine ischemic stroke models[45]. Future studies should specifically investigate the cellular sources of IL-17A in CPB-associated neuroinflammation.”